# Intimate Relationships as Perceived by Adolescents: Concepts and Meanings

**DOI:** 10.3390/ijerph18052256

**Published:** 2021-02-25

**Authors:** Isabel Moreira, Maria Fernandes, Armando Silva, Cristina Veríssimo, Maria Leitão, Luísa Filipe, Maria Sá

**Affiliations:** 1Unidade de Investigação em Ciências da Saúde: Enfermagem, 3004-011 Coimbra, Portugal; isabelf@esenfc.pt (M.F.); armandos@esenfc.pt (A.S.); cristina@esenfc.pt (C.V.); mneto@esenfc.pt (M.L.); 2Escola Superior de Enfermagem de Coimbra, 3004-011 Coimbra, Portugal

**Keywords:** intimate relationships, adolescent, dating, friends with benefits, primary prevention, health promotion

## Abstract

Adolescence is a period of great changes and the assumption of risk behaviours at the level of sexuality may have implications for health and well-being. Nowadays, adolescents live free from constraints and prioritise freedom, using their own terminology to label their relationships, it becoming in turn important to conceptualise intimacy relationships from their perspective. Therefore, a qualitative, descriptive, and exploratory study was performed. Participants included 109 adolescents aged 14 and 18 years old from public schools in central Portugal. Data were collected using 12 focus groups and a content analysis was undertaken. These terms attributed to intimate relationships by adolescents are, for the most part, mutual for both genders: crush, friendzone, friends with benefits, making out, dating, and similar in terms of meaning. In an intimate relationship, adolescents give priority to factors such as respect, trust, and love. The fear of loneliness, obsession, and low self-esteem are reasons pointed out by adolescents for maintaining an unhealthy intimate relationship. Adolescents’ knowledge of language about their intimate relationships is essential to establish effective communication and to build intervention programs in the healthy intimacy relationships field.

## 1. Introduction

Demographic changes over the last two centuries have led to adolescents currently living with social openness and accepting the emergence of new intimate relationship types. In this period of adolescents’ lives, they open up to their friends and begin to create relationships outside the family sphere, and they develop values and behaviours where sexual impulses emerge [1].

Currently with the “strong process of individualisation”, adolescents prioritise their freedom, leading them to “enjoy life more, which means living in search of more and greater pleasures, free from any limitations or constraints that may be imposed by a fixed loving partner” [2] (p. 41). This cultural change has led adolescents to develop new terminologies to describe their relationships [3]. Authors mention that they use a rich vocabulary to describe their relationships, translating the variation in intensity, expectations, and behaviours in these relationships. The meaning attributed to intimate relationships is better understood from the position of the interveners themselves. The decoding of the terms in use is a first step to understanding the meaning they give to the relationship, and how they build it and experience it. Greater knowledge of the terminology used by adolescents is highlighted given that it is important to learn what behaviours they demonstrate in these relationships.

Adolescence is a period of countless physical, emotional, and social changes. It is also a time of risk-taking and peer pressure. Attitudes and values related to gender equality, sexuality, and health influence the behaviours established in this period with strong implications at a later stage of adolescents’ lives, on their health and well-being, as well as their psychological and social development. Furthermore, this phase of life is also characterised by being the moment when they begin the exploitation of their sexuality, develop intimate relationships with other people, and start sexual activity [4].

Thus, adolescence is a critical moment and it can be understood as a window of opportunity to develop healthy behaviours concerning sex education and the prevention of violence in intimate relationships [5].

Adolescence is the stage in which adolescents seek autonomy and self-knowledge, initiate sexual relations, and there is a progressive distancing from the sources of support, namely, from the family, so that the study of sex-affective relationships in adolescents is essential. [6]. Many adolescents are already sexually active and some present problems resulting from early sexual activity, demonstrating a lack of control, particularly on when, where, with whom, and how to have sex. Male adolescents are more often persuaded into sexual practices and to do so unprotected, compared to adults [7].

Nowadays, different types of relationship between adolescents are guided by the softening of norms and rules, self-satisfaction, self-realisation, the experience of the moment, freedom, and greater pragmatism [2]. The concept of an intimate relationship is characterised according to Bradbury [8] as a particularly close interpersonal relationship. It is often used to define a sexual relationship. However, it can also be used to describe a relationship that does not include the sexual dimension.

Recent studies conducted in the United States regarding intimacy relationships present a greater acceptance of casual sex and a greater number of non-romantic sexual partners [9]. However, given the preponderant role of culture in the construction of relationships, this reality is not transversal to all populations.

As Sternberg points out, components present in a loving relationship—passion, intimacy, and commitment—are lived differently by the elements involved in the relationship, being modelled by socially learned roles. In light of the theory developed by the author, the three components mentioned above form the vertices of a triangle interacting with each other and forming different types of loving experience. This theory provides a basis for understanding many aspects of the underlying love in intimate relationships [10].

The study by Lavoie and co-workers found that one in five adolescents (22%) reported having a friend with benefits (FB) in the last 12 months and 36% had the intention of having one in the next three months [11]. Of the adolescents who reported having an FB, approximately half had one and the remainder had between two, three, or more FB. Most of them (97%) experienced genital touching and between 49% and 63% had genital or oral sex. For most of the participants, the first relationship with an FB occurred at age 15 (72%), with a person of the opposite sex (99%). Considering the results presented in the literature, it is important to continue to produce knowledge about this phenomenon.

A study conducted in 2020, in Portugal, regarding intimate relationships, with 4598 adolescents ranging from 14 to 17 years old, states that 67% of its participants do not acknowledge the control, stalking and pressure to have sexual relations as violent behaviour. In this study, it is also stated that the male contributes 4 times more pressure to have sexual relations than his partner [12].

In this way, it is necessary to understand the uniqueness and continuity of the first intimate relationships and to explore additional features to help adolescents improve communication, between peers, so that the relationship becomes healthier as Kan argues [13].

Parents, as well as society, in a general way, do not take the relationships between adolescents seriously. Although short and transitory, these experiences have enormous importance in adolescents’ lives. The experiences lived in these relationships not only affect their self-esteem, but they also constitute a risk factor for depression and anxiety and in the way they build long-term relationships [14].

The results of a systematic review conducted by Gómez-López and co-workers [15] indicate that the quality of relationships, the history of shared experiences, the feeling of attachment, and the beliefs lived are recognised as modulators of behaviour in relationships and the well-being of the individuals in their adulthood.

Moreover, the research accomplished by Guzman and co-workers [3] emphasises that teen relationship patterns have potential implications for the reproductive health and well-being of adolescents and adults. Also, Davila and co-workers [16] argue that the promotion of healthy relationships should begin in adolescence, before starting a relationship with commitment. This is not only to foster the adjustment between the elements of the dyad but also to minimize the problems that arise from a dysfunctional relationship. Program implementation in adulthood may be, according to those authors, one of the main reasons for the low success rate of the unhealthy relationship prevention programs among adults [16].

Nowadays, intervention programs are already applied in our society and are intended to promote healthy relationships. This kind of relationship is characterised according to eight fundamental principles of which respect, honesty, equality, safety, and physical and sexual respect stand out. Within a healthy relationship, a dependency on the other element of the relationship should not be verified. Partners treat each other as they would like to be treated by accepting their friendships, interests, and opinions (principle of respect). The honesty that should be in place between the couple, leads to the sharing of opinions, feelings, as well as all crucial facts for both parties. Responsibility sharing and also decisions should exist and be fair in such a way that there is equality in the relationship. In a relationship, stakeholders should feel secure not only in the surrounding area but also concerning their physical integrity. There should be additional respect for sexual practice, and for the fact that it should not occur in a forced manner or when the intervening party does not feel comfortable doing so [17].

The purpose of this investigation is to conceptualise intimate relationships from the perspective of adolescents, with the objectives of understanding the concepts in use, identifying the components of healthy intimate relationships, and identifying the reasons that lead adolescents to maintain unhealthy intimate relationships.

## 2. Materials and Methods

A descriptive and exploratory study with a qualitative approach was performed, integrating a more comprehensive validation investigation of a Healthy Intimacy Relations Promotion Program (PRIS), starting in 2016. Participants comprised 109 adolescents aged between 14 and 18 years old, attending primary and secondary education in schools in the central region of Portugal. It was verified that in terms of gender representation, the sample’s composition confirmed a similar representation. Of the 109 participating adolescents in this study, 57 (52.3%) were female and 52 (47.7%) were male. Regarding the adolescents’ age, the most representative groups corresponded to those 16 years old (60.5%). Participants’ minimal age varied from a minimum of 14 while the highest age was 18 (Table 1).

The sample was established according to Poupart [18], in order to have a global and in-depth picture of the concepts that adolescents who attend that class’s level of education have about intimate relationships.

Data collection was performed in 12 focus groups (FG), using an interview script.

This data selection technique was justified in a way that is recommended in the exploratory phase of a research project [19]. It allows interaction between participants and highlights their opinions, experiences, as well as the language they use [20]. This study was part of a validation study of a program to promote healthy intimate relationships concerning the development of adolescents’ awareness about violence in intimate relationships, empowering them to build healthy intimate relationships. The program consisted of 9 sessions, during two school sessions of 90 min each. A focus group was held in the first session from which we began to work to promote healthy intimate relationships with adolescents. The main goal of this session was to work through the concept of intimate relationships between adolescents concerning the risk and protective factors of a relationship. During some focus groups, adolescents’ teachers remained in the classroom while the training session was held.

Each focus group was conducted by two members of the research team in which one assumed the role of interviewer and the other the role of an observer. A nursing degree course student was also integrated into the research, as a facilitator of the interaction between participants due to the proximity established based on their ages and languages. Conduct of the focus group was flexible according to the interaction established between participants, obeying the use of a set of topics that were part of a semi-structured interview script (Table 2).

At the beginning of each focus group, the anonymity of the participants was guaranteed through the assigning of a number and letter to each focus group and requesting their participation. The number of adolescents in each focus group varied between 4 and 12, depending on the number of students per class, having 6 female adolescents and 6 male adolescents. The focus group average time was 30 to 40 min. Some of the adolescents found initial inhibition to talk about the subject and often there were some non-verbal behaviours (e.g., gaze deviation, laughter, etc.) revealing their inhibition, particularly in groups constituted by boys. The collected data were handwritten during the focus group and later analysed according to the assumptions defined for content analysis by Poupart [18].

Thus, this procedure began with: data reading and full rereading in order to perform a thorough examination; lexical relationships between data were searched, writing notes and codes; an integrating concept was assigned that corresponds to a category. In this procedure, four themes: terms and meanings attributed to intimacy relationships; factors that condition a relationship establishment; components of a healthy relationship and reasons to remain in an unhealthy relationship. Within the theme factors that condition a relationship establishment, categories emerged: attraction, affection, and sexual pleasure, which allowed factors’ explanation that leads to the establishment of a relationship. This was an inductive process but at the same time the terms were identified by sex, and the indexes of each term were defined. At the end of this process, the analysis performed by researchers was given to a group of adolescents for results of validation, as recommended by these authors. At the beginning of the study, a meeting was held with the parents in charge of education to explain the scope and nature of the study, to request the participation of their students, and to obtain their informed, free and explicit consent. Consequently, access to adolescents was guaranteed by teachers who joined the program (PRIS). The study received the assent of the Ethics Committee of UICISA: E No. 297-08-2015.

## 3. Results

Multiple reading of the analytical corpus brought out the following themes: terms and meanings attributed to intimacy relationships; factors that condition a relationship establishment; components of a healthy relationship; and reasons to remain in an unhealthy relationship.

### 3.1. Concepts and Meanings

Regarding the terms and meanings attributed to intimate relationships, the majority are common to boys and girls, as can be seen in Table 3. A similar situation occurs with the meanings they use to explain them.

The term crush is used by male adolescents to refer to the attraction (physical) and desire they feel for another person. Crush is “*having a weakness for someone, an attraction, an interest*” (FG2), or “*a physical attraction to someone we know, personally or not!*” (FG1). The term to fancy someone is understood by male adolescents as having a relationship with someone without commitment or responsibility, or as they say, is “*having physical and non-emotional involvement*” (FG4).

When they use the word making out, they report to a relationship based on physical pleasure without commitment, mentioning, as an example, that “*it is when a couple kisses and has sex and the relationship is not assumed*” (FG3). Accordingly, the term making out is also used by girls to define this type of relationship.

When referring to sexual relations between friends, adolescents use the expression friends with benefits, as they allude to are the “*friends who have physical involvement, having the same intentions*” (FG2) or those “*who practice the act of dry hump regularly, but do not have a loving relationship with each other*” (FG6). This becomes a purely physical relationship, without commitment, that is not assumed, as expressed in the adolescents’ report “*it is essentially physical, without emotional investment, which can evolve into dating*” (FG5).

Regarding the relationship context between friends, they use the expression of colourful friends to refer to close friends for whom they are attracted.

From the adolescents’ storytelling, the term friendzone has also emerged, being used to characterise relationships between two people who like each other. However, this kind of relationship becomes distinct once one of the elements intends more than friendship and the other does not, placing it in the friendzone (on social networks), as expressed in the following statement:

“*When someone likes us, but the feeling is not reciprocal, then we put them in the friendzone*”.(FG1)

In the lexicon of the terms used by male adolescents to refer to a purely sexual relationship, on the part of the girls the terms “one-night stand” emerged and on the part of the boys to shag, hump or quickie.

When they use the term hang out, they are referring to the beginning of an affective relationship, which for some still has no commitment and for others is “*early-stage dating*” (FG2), an “*unofficial, uncommitted dating*” (FG4). The term dating means for male adolescents a “serious” relationship, in which there is love, and also emotional and physical involvement. It is also associated with a commitment from the two elements involved in the relationship. These relationships are differentiated from the others by commitment, seriousness, fidelity, and longer duration. They are considered the “last step” of the journey of an intimate relationship.

“*Mutual love relationship with commitments from part to part. Involves a greater knowledge of the other person. Relationship of respect, more serious relationship than the previous ones [crush, colourful friends, making out...]*”.(FG1)

### 3.2. Factors for a Relationship Establishment

In the configuration of the terms attributed by adolescents to intimacy relationships, three factors emerged for the establishment of a relationship: attraction; sexual satisfaction/sexual pleasure, according to Table 4.

The subcategories that compose the attraction category are associated with the interest in each other making them feel a crush for the other. The attraction is essentially linked to the physical traits of the person, it may even be unsure that the other has an interest in him, as illustrated in his speeches:

“*physical attraction that may be for an unknown person—the other person does not even know*”.(FG2)

When they refer to relationships called: friendzone, to go out with/to hang out, colourful friends, dating, the affection emerges such as the feeling/emotion that surrounds them. However, the friendzone definition denotes some variability due to the feeling/emotion manifestation that is not felt in the same way by both relationship elements. While one feels this affection as friendship, the other expresses it as being love:

“*Two people who like each other, however only one wants more than friendship*”.(FG2)

When they refer to dating they speak of love, of a deep feeling. It is an affective relationship in which feelings are shared, there are emotional involvement, and a mutual bond. Therefore, it is affection and love that lead to compromise:

“*Making a commitment to be faithful, there is an only commitment when there is love*”.(FG2)

Relationships that integrate the category of sexual pleasure, called making out, dry hump, sex and to fancy someone, are characterised by physical involvement in which there is no emotional involvement and no commitment between the elements involved in the relationship. It is a short-period, a fleeting relationship, that can be “*one day or more than a day*” (FG2), and which is essentially focused on the consummation and satisfaction of sexual desires.

From the analysis of the terms used by adolescents to describe the intimacy relationships, a variation in commitment emerges, according to the type of relationship and the involvement of adolescents in it.

Thus, in relationships in which they seek sexual pleasure, there is no affective involvement, momentary relationships, or a short (duration) relationship characterised by the non-commitment between the elements involved in the relationship. On the other hand, in affective relationships in which there is emotional involvement and the construction of a sense of attachment, there is a commitment as can be seen in the spectrum requested by adolescents regarding relationships and commitment (Figure 1).

Similar to the representation of commitment variation in intimacy relationships between adolescents, a broad spectrum concerning intimacy has been performed. This involves the sharing of the inner and outer self, a self-revelation as it is conceived in the expression “*involves the things that we only share with people … with which we feel comfortable to say or show what we should not show others at first*” (FG5), to expose the “*insecurity and personal things to the other*” (FG4). Regarding the adolescents, for them to have intimacy there must be trust in the other person, as well as respect to feel at ease once “*intimacy is the part of the private relationship with the two people*” (FG5). Adolescents report that there is no intimacy in a crush, while it can already be perceived and expressed among colourful friends and is present when referring to dating. It is also mentioned that “*in friends with benefits there is only physical intimacy, while in other relationships there is emotional intimacy*” (FG5). According to them, this component of intimate relationships depends on the type of relationship (Figure 2).

Love relationships can be characterised where adolescents of both sexes highlight passion as another significant and differentiating component in relationships. They describe passion primarily as an “*intense and temporary feeling that can be the beginning of a loving relationship*” (FG5). It is considered as an irrational behaviour that they “*cannot understand, nor explain*” or “*is an intense physical and psychological attraction*” (FG6), which can be experienced momentarily or can last for some time, and may evolve into love. They assume that in relationships that are based on sexual pleasure, namely on a crush and on making out there is no passion, there is “*attraction and not feeling*”, and these are “*for physical satisfaction*” (FG6) (Figure 3).

### 3.3. Components of Healthy Relationships

Regarding the components of a healthy relationship, male adolescents, points out: respect, trust, and love. Female adolescents present exclusively loyalty/fidelity. Communication, caring/affection, understanding, and friendship are valued both by boys and girls, although they are more referenced by girls. On the other hand, boys value honesty and support/availability/help, more than girls do. Sexual intimacy/sexual relations are referred to only by male adolescents, as well as the duration of the relationship, even if short, as can be seen in Table 5.

The reasons highlighted by male adolescents to stay in an unhealthy relationship, sometimes refer to intimate relationships in general, not focusing exclusively on adolescence. Most of these reasons are different for male and female adolescents, as can be seen in Table 6. The main reason pointed out by adolescents of both genders is the fear of being alone. Girls also point as main reasons: group pressure (“all have a boyfriend minus them”), the insecurity, the hope of changing the other element and the recognition that love makes them go blind, not able to see the error. For boys, one of the main reasons is based on the economic aspects. Both boys and girls highlight low self-esteem and obsession as possible reasons to stay in an unhealthy relationship. Boys also mentioned the existence of children, the feeling of pity, despair, jealousy and difficulty in delimiting what is an unhealthy relationship. By female adolescents was mentioned the legitimation of control on the part of the other, considering that the control and feeling that someone likes them justifies remaining in the relationship.

## 4. Discussion

Adolescents, generally use an extremely rich, diverse, and meaningful vocabulary to describe their intimate relationships. Most of the terms used are similar in the language applied by boys and girls. They are also similar in terms of the meanings attributed to the used terms such as: to fancy someone, friends with benefits, crush, among others.

The extended lexicon of terms used by adolescents of both genders to express their relationships may be related to the evolution of semantics. This becomes the result of social changes that have occurred throughout history and the need to introduce communication solutions that increase the likelihood of communication success among adolescents since the previous semantics did not allow it [21].

This widening of semantics is global. From the analysis of the literature, it is verified that the concepts used by adolescents are quite often cross-cultural concepts. “Friends with benefits” that Lavoie and co-workers conceptualize as “a sexual experience with a friend, without the expectation of a long-term relationship or romantic commitment” is an example of a similar situation that was found in the present investigation [11].

A relationship characterisation analysis of terms denotes that many of the terms used by adolescents refer to a fleeting relationship, of a short time duration, where there is only physical involvement and no commitment between the elements during the relationship. Williams and Russell also described how new forms of sexual relationship challenged the idea that sexual relations between adolescents occur mainly in the context of dating or romance, and considered that they occur in casual relationships between partners. This could be a worrying situation by increasing the risks associated with occasional relationships, the possible occurrence of sexually transmitted diseases, and future problems related to their sexual and reproductive health [22].

Guzman’s research [3] reports that boys consider their partners as disposable in less serious relationships, without a commitment. In the same way, it is important to highlight the significant impact that gender-based social norms have on boys’ and girls’ choices. Gender inequality influences sexuality in its expression and behaviour. In many social environments, adolescents have a low level of power or control in their sexual relations. They may be unable to negotiate sexual activity with their partners, especially if they are in a sexual relationship with older men or in relationships involving exchanging sex for money or gifts [23].

The experience of relationships based on satisfaction and physical and/or sexual pleasure can be explained according to Chaves [2] by the characteristics of today’s society. Among these, the importance given to the here and now, losing the meaning of a long-term vision, can be emphasised by the ease of building essentially utilitarian relationships. The value given to individual freedom leads to a search for the satisfaction of their own needs and private happiness.

Adolescents distinguish dating from other relationships. This becomes the result of the understanding of affective and emotional involvement, of longer duration, a commitment of both partners, and the taking over of the relationship with others. Although these components are not present in other types of relationships, it is in dating that adolescents refer to higher levels of passion, that there can be found greater intimacy, and the construction of a sense of bonding with each other. There is the establishment of a commitment, which is in accordance with the theory of the love triangle presented by Sternberg [10]. Despite this author being exclusively focused on love relationships, it can be said that in the present investigation adolescents, when referring to the relationships they entitled crush, to fancy someone and making out, point out a lack of passion, commitment, and intimacy. Considering the other types of relationships such as a friend with benefits, these comprise physical intimacy, the passion that can be lived briefly or for short periods of time, and with low commitment. The study by Bisson and Levine [24] presented similar results when it referred to friends with benefits, given that the relationship has levels of passion consistent with romantic relationships, but lacks the characteristic commitment of these relationships.

Adolescents, when asked what they thought about healthy relationships consider that for the establishment of healthy relationships, respect and trust are fundamental values. Moreover, Guzman and co-workers [3] concluded that adolescents attributed respect to be an essential value for building a healthy and successful relationship. Costa and Modesto’s work [25] concerning the social representation of healthy love relationships also points out that respect becomes the core element in a healthy loving relationship. On the other hand, Smeha and Oliveira [26] point out that lack of respect is one of the aspects that makes it difficult to establish love relationships in young adults.

Considering this valorisation scale, love, loyalty, communication, affection/affection, understanding, and cooperation emerge as other relevant components. It stands out that loyalty is only identified by female adolescents, while male adolescents emphasise sexual intimacy and sexual intercourse. Similarly, Williams and Russell [22] report that adolescents with fixed partners show more care, they are more cooperative, and communicate more with each other. Taking into account the number of adolescents who refer to the components of communication, tolerance/patience/know how to deal with the other and understanding as important in a healthy relationship in the light of what Blais reported [21], it seems to be possible to consider that those girls are more aware than boys of the investment needed to maintain a relationship. An explanatory possibility that only male adolescents present the idea that the sexual dimension is important in a healthy relationship can be supported by Baptista’s argument [27] when he mentions that boys, especially younger adolescents, are those who most desire and practice sexual relations without emotional involvement. This dimension of the relationship was also confirmed in a study developed by UMAR (Women’s Association Alternative and Response) in 2020 in which boys pushed harder than girls to have sex [12]. Considering the justifications presented about maintaining an unhealthy intimate relationship, the fear of being alone, obsession, and low self-esteem arises in boys and girls. Female adolescents have a specific vision associated with romantic love. In this way, they reproduce characteristics associated with romantic love as reasons for staying in an unhealthy relationship. Sánchez-Sicilia and Serra also identified this connection, transversal to all adolescents, in their research on their love discourse [6].

The reasons highlighted by adolescents to remain in an unhealthy intimate relationship, despite the knowledge they reveal about healthy intimate relationships, show that their decision is based on elements that go beyond knowledge. The decision to remain in an intimate relationship that is not healthy or safe reinforces the importance of personal development for decision-making.

However, in this work some limitations were identified in Portugal, related to the reduced recent research on the topic. This did not allow us to have a reference to confront these results with cultural specificity. Despite the richness and detail of the information obtained in some interviews, contextual factors, in particular since the interviews have taken place in the classroom with the classroom teacher presence. This may have influenced the adolescents in expressing their ideas and explaining the meanings they attribute to them. However, it is questioned whether these contextual constraints have not been minimised by the presence of a nursing student, as an observer, with an age close to the elements of the groups and resorting to focus group interviews, as this was a strategy that facilitated more genuine discourses.

## 5. Conclusions

Adolescents use their language codes to differentiate the multiple hues of their intimate relationships, ranging from short to long-lasting relationships, with different levels of commitment, often based on individual, momentary satisfaction. The conceptualisation and consequent understanding of the intimate relationships experienced by adolescents are essential to prevent violence and promote healthy intimate relationships. These aspects are configured as a first step towards more strategic and coherent interventions with the adolescents’ conceptions and experiences. Nurses, especially those who work more directly with the community and these age groups, present themselves as important agents of debate and intervention, building actions aimed at preventing violence in intimate relationships and, in particular, for the promotion of adolescents’ health. Despite the limitations, these results may make relevant contributions to the construction of programs aimed at the personal and social development of adolescents and the construction of a program to help parents and educators. It is important that health professionals work with adolescents on their risks in intimate relationships with a focus on pleasure, since these risks can have repercussions on their physical and emotional health. It is also recognised that behaviours developed during adolescence have a lifelong impact and that intervention at this stage of life benefits all individuals and can contribute to a healthier development and more equitable societies. Therefore, it should be noted that interventions to promote healthy relationships should occur early, before a relationship with commitment. It is also understood that this investigation contributes to revealing of the terminology used by adolescents currently, which allows them to be approached. It offers an important contribution to the knowledge of adolescent behaviours, fundamental elements for establishing healthy intimate relationships, and the reasons for staying in an unhealthy relationship. These potentially aid the planning of preventive interventions in the school and health contexts. It becomes also important, however, to understand how the experience of different types of relationship has implications for the well-being of adolescents and in the construction of affective relationships in adulthood, because most studies have focused on unhealthy intimacy relationships. Even so, is important to examine how the different types of relationship evolve as this is a relevant aspect even if it has not been explored in this study. Considering the elements that characterize a healthy relationship whether it is casual or committed, further studies should include. larger, more heterogeneous samples with adolescents from different cultures and religions to understand how these dimensions condition relationships.

## Figures and Tables

**Figure 1 ijerph-18-02256-f001:**
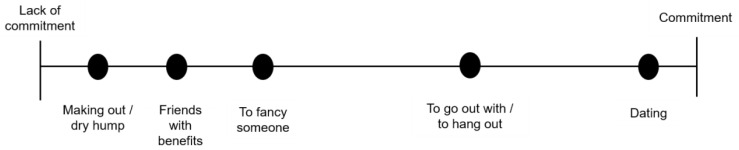
Spectrum of the representation of commitment in love relationships between adolescents.

**Figure 2 ijerph-18-02256-f002:**
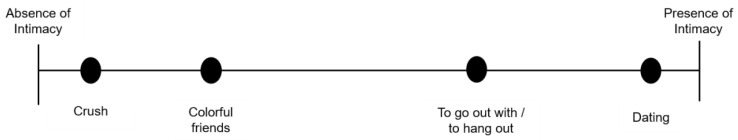
Spectrum of the representation of intimacy in love relationships between adolescents.

**Figure 3 ijerph-18-02256-f003:**
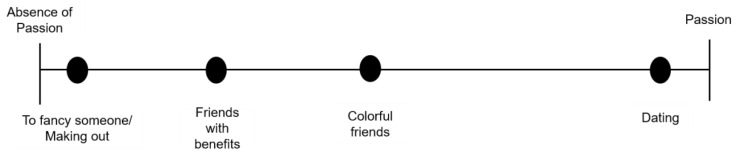
Spectrum of the representation of passion in love relationships between adolescents.

**Table 1 ijerph-18-02256-t001:** Distribution of absolute (*n*) and relative frequencies (%) from adolescents according to their gender and age.

	Global (*n* = 109)	
	*n*	%
**Gender**		
Female	57	52.3
Male	52	47.7
**Age (years)**		
14	1	0.9
15	6	5.5
16	66	60.5
17	32	29.4
18	4	3.7

**Table 2 ijerph-18-02256-t002:** Focus group guiding topics.

Guide Topics for the Focus Group
What terms or expressions do you use to characterise your intimate relationships? What meaning(s) is attributed to each term or expression? What elements constitute a healthy relationship. How would you place them in a continuum between absence and presence in each of the described relationships? What makes you maintain a relationship that you consider unhealthy?

**Table 3 ijerph-18-02256-t003:** Terms attributed by adolescents (boys and girls) to intimate relationships.

Girls	Boys
Crush	Crush
To fancy someone	To fancy someone
Making out/dry hump	Making out
Friends with benefits	Friends with benefits
Colourful friends	Colourful friends
Friendzone	Friendzone
One night stand	To shag/hump/quickie
To go out with/To hang out	To go out with/To hang out
Dating	Dating

**Table 4 ijerph-18-02256-t004:** Factors for establishing a relationship.

Category	Subcategory
Attraction	Crush
	Colourful friends
Affection	Friendzone
	Colourful friends
	To go out with/to hang out
	Dating
Sexual pleasure	Friends with benefits
	Making out/dry hump
	Having sex
	To fancy someone

**Table 5 ijerph-18-02256-t005:** Components of a healthy relationship.

Components	Girls	Boys
	*n*	*n*
Respect	19	13
Trust	15	9
Love	10	7
Loyalty	11	-
Communication	9	5
Tolerance/Patience/Knowing how to deal with others	6	1
Understanding	6	3
Affection	6	4
Cooperation	6	3
Friendship	7	1
Knowing each other and ourselves	3	2
Honesty	3	5
Empathy	3	-
Freedom	2	3
Support/Availability/Help	2	5
Commitment	2	1
Sexual intimacy/intercourse	-	6
Time (for relationship)	-	3
Equality	1	3

**Table 6 ijerph-18-02256-t006:** Reasons why adolescents remain in an unhealthy relationship.

Reasons	Girls	Boys
	*n*	*n*
Fear of being alone	4	2
Love is blind, we do not notice error	3	-
The hope of changing the other	2	-
Insecurity	2	-
Group pressure	2	-
Economic aspects	-	2
Obsession	1	1
Low self esteem	1	1
Children	-	1
Pity	-	1
Despair	-	1
Jealousy	-	1
Difficulty knowing what is right in a relationship	-	1
Feel that someone likes us	1	-

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
