# Peer review of "Intimate Relationships as Perceived by Adolescents: Concepts and Meanings"

_ijerph, 2021, doi:10.3390/ijerph18052256_

Round 1
Reviewer 1 Report
Review of: Intimate relationships as perceived by adolescents: concepts 2 and meanings
Comments to the Author
This manuscript is an interesting topic.
The article in its current form need more rewriting, clearer structure, and more concrete analyses of the data.
I have some, major issues, which I would like the authors to, respond to, as well as number of mostly minor comments which could improve the Ms.
MAJOR ISSUES
The major issues concern.
Introduction:
It would be useful to expand the intro with more references and fact on the general picture of sexual debut and behavior in Portugal among young people. It will give a reflection to the results that are outlined in the article. Insert more references in the intro on, what does the existing knowledge say and how does this article contribute with new knowledge? These dimensions are a bit unclear and should be addressed.
As a reader more references will reinforce the desire to read the article closer and strengthen the discussion about this issue.
It is recommended to clarify what the research question is - it may be in a separate section.
Materials and Methods:
The analysis strategy is unclear. How is the large data material analyzed, based on which scientific approach, and based on which coding?
The material is based on 109 participants and therefore it may come as a surprise that the large material has not been utilized better in the actual reporting of the results. As a reader, you miss direct sections from the focus group interviews - how the dynamics emerge and what social practices are formed in the groups.
One of the great advantages of focus group - interview is that this is focused, that there is talk of a specific topic, which is determined by the researcher, and that the focal point is the interaction that takes place. One can also say that the social interaction or group dynamics is the tool to achieve the desired knowledge of the social phenomenon that is in focus. (Morgan 1997). The focus group interview is particularly suitable for analyzing and examining how opinions, attitudes and identity are formed in social groups. There sections from the interviews are of high interest. It is the group interaction itself that produces the data (Morgan 1997).
Another factor that is underlined in the article now is the gender dimension, it will be extremely interesting to analyze this, how the girls and boys differ - do they always or only in parts of the groups, why / why not.
Furthermore, a reflection on, advantages and disadvantages of this method and why one has chosen to report the analyzes in a more quantitative form rather than qualitative with quotations and sections of statements is missing.
Finally clarify who has read the material in the different phases - which authors have contributed to the analyses?
Discussion: The discussion is ok. I think it can be strengthened by what is new knowledge and what research is needed in the future.
Limitations: There is a need for further reflection on shortcomings and limitations in the study in general
Minor
* It would be good to have extra tables of the study – participants, coding/ themes etc.
* How does religion, more restrictive sexual norms affect young people and is there a difference in the material?
Author Response
We want to express our gratitude for your valuable recommendations towards the improvement of the manuscript. We have performed changes accordingly, which we believe improved significantly the quality of this manuscript.
Recommendation "It would be useful to expand the intro with more references and fact on the general picture of sexual debut and behavior in Portugal among young people. It will give a reflection to the results that are outlined in the article."
Justification and amendments: We agree that the absence of references to the portuguese context might be seen as a limitation, but it also reflects the study's novelty. Following your suggestions, we sought further evidence on this scientific domain, but we were not able to identify any more research products. Although we could pursue a comparative analysis with evidence from other countries, the cultural meaning would be missing.
Recommendation: "Insert more references in the intro on, what does the existing knowledge say and how does this article contribute with new knowledge? These dimensions are a bit unclear and should be addressed. As a reader more references will reinforce the desire to read the article closer and strengthen the discussion about this issue."
Justification and amendments: We have integrated new references to support all the work mainly across the introduction and discussion sections.
Recommendation: "It is recommended to clarify what the research question is - it may be in a separate section"
Justification and amendments: With regards to the reviewer recommendations concerning the research question, we have rewritten it in a more explicit way and placed it in a separate paragraph at the end of the introduction.
Recommendation:
- a) The analysis strategy is unclear. How is the large data material analyzed, based on which scientific approach, and based on which coding? The material is based on 109 participants and therefore it may come as a surprise that the large material has not been utilized better in the actual reporting of the results. As a reader, you miss direct sections from the focus group interviews - how the dynamics emerge and what social practices are formed in the groups.
b) It would be good to have extra tables of the study – participants, coding/ themes etc.
Justification and amendments: We have included tables with participants' data, as well as, the interview script and we do agree that these amendments enhance greatly the manuscript comprehensibility. We have re-written the research procedures and elaborate them in a way that we believe allow for a better perception of how the data presented was collected and analysed.
Recommendation: "Another factor that is underlined in the article now is the gender dimension, it will be extremely interesting to analyze this, how the girls and boys differ - do they always or only in parts of the groups, why / why not."
Justification and amendments: We also have considered the suggestion of analysis of how boys and girls differ, concerning meanings and concepts differences are noticeable between boys’ and girls’ answers.
Recommendation: There is a need for further reflection on shortcomings and limitations in the study in general
Justification and amendments: Regarding the discussion and study limitations, these were clarified justifying the scarce research in the area to be developed in Portugal, as well as factors such as the elements present in the room where the focus groups occurred that may have influenced the adolescents' responses. It is also suggested to carry out new studies incorporating larger samples both in terms of the number of participants, different cultures, and religions using heterogeneous samples to verify how these factors have or do not influence relationships.
Recommendation: "Finally clarify who has read the material in the different phases - which authors have contributed to the analyses?"
Justification and amendments: Taking into account another item, all the authors read the manuscript in different phases. Some of the authors were responsible for data collection (I.M.; M.S.; C.V.; M.F.; A.S.) while others were responsible for the presented data analysis in the manuscript (I.M.; M.S.; M.F.). The work of writing - original draft preparation was performed by I.M.; M.S.; M.F.; L.F., while the final revision of the manuscript and its edition was at the responsibility of other authors (M.F.; M.L.; L.F.).
Reviewer 2 Report
This paper presents the findings from qualitative research with adolescents about how they conceptualise their intimate relationships.
The introductory/literature review sections sets up the focus of the paper and emphasises the importance of the paper - it argues that intimate relationships in adolescence have consequences both at the time and for the future health and wellbeing of individuals, and that it is important to understand adolescents' perspectives on the meanings of intimate relationships. I'm not sure if the structuring of the discussion is typical for this paper, but there are some quite short paragraphs here and the points thus appear quite brief. It would perhaps be good to see some of the points fleshed out a bit more. More substantively, the authors mention 'healthy relationships' at the end of this section when outlining the focus of the paper. It seems that they are positioning healthy relationships as an objective outcome/reality in themselves rather than something that is also socially constructed. It would perhaps be worth seeing the authors discuss this concept in more depth. Further, it is briefly noted that 'values related to gender equality' may shape experiences in adolescence. Some more awareness of the socio-cultural-structural influences on adolescent relationships would, in general, be good to see. Some of the statements made about adolescence and the changing patterns of relationships are made with some firmness when it fact they may socially and structurally contingent.
Some further detail on the methods is necessary. What is the program that is referenced to here? What does it exactly involve? When/where were the focus groups conducted? Were they part of the program or afterward? Is there any further data on the participants (sexuality, ethnicity...)? What exactly were participants asked about in the groups? Was there a focus group guide? Who conducted/moderated the groups? And how was the data analysed? Some explanation of the analytical approach/technique would be good to see. Also - to what end were these focus groups conducted? Was it part of a wider study/evaluation of the program? What was the authors' involvement with the program? Finally, how were research ethics managed? Given that there could have been disclosures made during the groups?
The quotes in the data analysis are presented with just the information about the groups. Who said these quotes? Boys, girls? What age were they? Were there differences in language among participants?
Some quotes are not clear - what is meant by 'friends who eat themselves'? (line 153) or 'colored - or is it meant to be colourful? - friends' (p158)
There is some greater depth in the part about healthy relationships in terms of gender differences.
The discussion pulls out how adolescent relationships are becoming more diverse in form and function, and the conclusion notes that it is important to consider pleasure here. It may be worth exploring the extent to which adolescent relationships are pathologised by adults around them and in wider public discourse, and the value of taking an adolescent-centred approach.
That being said, I think there is a need to more fully bring to life the analysis. The data is presented quite uniformly - other than the gender differences mentioned, there is little impression of how participants constructed their perspectives within the groups and the similarities/differences between groups and individuals. Likewise, there is a need for more context - how did these accounts emerge? What were the questions the participants were asked? What dialogue occurred between participants and the facilitators?
Overall, there is some interesting data here but there needs to be some more contextual information about how the data was generated and what emerged from the different groups, particularly given there is a wide age range here (14 to 18).
Re the English language editing required - while the piece is written clearly overall, there seems to be some translation errors here to pick up on.
Author Response
We want to express our gratitude for your valuable recommendations towards the improvement of the manuscript. We have performed changes accordingly, which we believe improved significantly the quality of this manuscript.
Recommendation: “I'm not sure if the structuring of the discussion is typical for this paper, but there are some quite short paragraphs here and the points thus appear quite brief. It would perhaps be good to see some of the points fleshed out a bit more.”
Justification and amendments: Regarding the received feedback, we were able to perform improvements, in which we highlight the development of important aspects present in the introduction as well as in results discussion, with the respective literature support.
Recommendation: “More substantively, the authors mention 'healthy relationships' at the end of this section when outlining the focus of the paper. It seems that they are positioning healthy relationships as an objective outcome/reality in themselves rather than something that is also socially constructed. It would perhaps be worth seeing the authors discuss this concept in more depth.”
Justification and amendments: Considering data discussion and according to the recommendation, we added and developed the basic points that characterize healthy intimacy relationships, in order to give more support and strength to the manuscript
Recommendation: “Some further detail on the methods is necessary. What is the program that is referenced to here? What does it exactly involve? When/where were the focus groups conducted? Were they part of the program or afterward? Is there any further data on the participants (sexuality, ethnicity...)? What exactly were participants asked about in the groups? Was there a focus group guide? Who conducted/moderated the groups? And how was the data analysed? Some explanation of the analytical approach/technique would be good to see. Also - to what end were these focus groups conducted? Was it part of a wider study/evaluation of the program? What was the authors' involvement with the program? Finally, how were research ethics managed? Given that there could have been disclosures made during the groups?”
Justification and amendments: Regarding the Intervention Program referred to in the article, PRIS program (Program for the Promotion of Healthy Intimate Relationships) is based on an intervention with the school community, consisting of 9 sessions (90 minutes each), developed during the adolescents' school time. The focus group, an integral part of the intervention, was held in the first session, from which the issues of promoting healthy intimacy relationships began to work with. Each focus group was conducted by 2 members of the research team (authors of the study) (1 interviewer element and 1 observer element), still with the collaboration and presence of a nursing student (as a facilitator of interaction with adolescents either by the age or by the language factor). The focus group followed a flexible conduction according to a set of topics that are part of the semi-structured interview script. Regarding the interview script, the following questions were asked: What terms or expressions do you use to characterize your intimate relationships? What meaning(s) is attributed to each term or expression? What elements constitute a healthy relationship. How would you place them in a continuum between absence and presence in each of the described relationships? What makes you maintain a relationship that you consider unhealthy? Regarding data analysis, after its collection they were analyzed according to the assumptions defined for content analysis described by Poupart. The procedure comprised: reading, full rereading of the data for a thorough examination; search for lexical data relations, writing notes and coding; attribution of an integrator concept corresponding to a category. After categories emerge, a relationship is established between them. This was an inductive process, with simultaneous identification of terms according to participants’ gender. After the analysis performed by the researchers, the results were validated by a group of adolescents. At the beginning of the study, a meeting was held with the parents in charge of education to explain the scope and nature of the study, request the participation of the students and obtain free and informed consent. Ethics issues are safeguarded as the study is approved by the Ethics Committee of UICISA:E No. 297-08-2015. All participants in the study signed free and informed consent. Regarding other questions raised about the participants and other factors that could have interference in relationships, such as in the focus group these ones were not addressed, such as ethnicity, sexuality, or even religion. This is one of the points that we present as a suggestion of future works, in which we highlight the need to use large, heterogeneous samples of different cultures and religions.
Recommendation: “Some quotes are not clear - what is meant by 'friends who eat themselves'? (line 153) or 'colored - or is it meant to be colourful? - friends' (p158)”
Justification and amendments: As for a very specific question about concepts "Friends who "eat" themselves" and "colored", they have already been corrected for "friends who have physical involvement" and "colourful", respectively.
Recommendation: “That being said, I think there is a need to more fully bring to life the analysis.”
Justification and amendments: Nonetheless, we believe that with all the performed changes in the manuscript we brought more momentum to the discussion itself.
Recommendation: “Re the English language editing required - while the piece is written clearly overall, there seems to be some translation errors here to pick up on.”
Justification and amendments: considering the recommendation given, we hereby declare that we have carried out a linguistic revision of the manuscript, in its full extension.
Round 2
Reviewer 1 Report
xxx good
Author Response
We want to express our gratitude due to the comment expressed related to our manuscript.
Reviewer 2 Report
Thank you for the opportunity to read the redrafted paper. I commend the authors for responding so constructively to the comments and I feel that the paper is in good shape.
Just a couple of suggestions that may or may not wish to be taken on board;
- The introduction is pretty comprehensive now. I just wonder whether it's worth considering the structure. In my view, it would be better to have the content related to the changing nature of adolescent relationships come first, then move on to the points about adolescence in general (i.e. about development and formation of the person), then the statement about relationships specifically being important at this time (notwithstanding tendencies in society to disregard them as trivial or fleeting) and then go into the implications of adolescent relationships e.g. in terms of health/wellbeing at the time as well as in the future. It can then finish on the point about intervention programmes and how these are better started early in life rather than waiting until adulthood. I think that would flow somewhat better and neatly set up the focus of the study in terms of investigating how adolescents are conceptualising their relationships at the current time and how they understand the concept of 'healthy relationships'. Of course this is just a suggestion and it may be decided that an alternative structure works just as well or better.
- Couple of points on the findings/discussion: First, were the different relationship types distinct points on the spectrum or could one lead to another over time? Perhaps this wasn't raised in the data but could be something for future research. Second, were participants' perspectives on 'healthy relationships' global? Or were the comments made vis-a-vis particular relationship types? What constitutes a healthy 'casual' relationship may be distinct from what constitutes a healthy 'committed' relationship? Again, if this wasn't explored then perhaps it's an avenue for future research.
Best wishes with the paper. I hope these suggestions are helpful.
Author Response
We want to express our gratitude for your valuable recommendations towards the improvement of the manuscript. We have performed changes accordingly, which we believe improved significantly the quality of this manuscript.
Recommendation: “The introduction is pretty comprehensive now. I just wonder whether it's worth considering the structure. In my view, it would be better to have the content related to the changing nature of adolescent relationships come first, then move on to the points about adolescence in general (…), then the statement about relationships specifically being important at this time (…) and then go into the implications of adolescent relationships e.g. in terms of health/wellbeing at the time as well as in the future. It can then finish on the point about intervention programmes (…).
Justification and amendments: The structure used in the initial version of the manuscript was based on a contextualization considering adolescence per se, as well as factors such as gender equality, sexuality, and behaviours observed in the individual development phase. Also, future implications considering health and well-being, adolescence as a stage in human development where sexual education and violence in intimate relationships prevention are explored, always taking into account literature studies. Moreover, the recommendation giver allowed us to see the guiding thread from the current manuscript introduction, since the changing nature of adolescent relationships to the intervention processes and the importance of these at younger ages, so the changes suggested by the reviewer were extremely helpful.
Recommendations: “First, were the different relationship types distinct points on the spectrum or could one lead to another over time? (…) Second, were participants' perspectives on 'healthy relationships' global? Or were the comments made vis-a-vis particular relationship types? What constitutes a healthy “casual” relationship may be distinct from what constitutes a healthy “committed' relationship?”
Justification and amendments: Although it is a theme that was not addressed in this study (line 553), it would be important to perform an analysis on how the different types of relationships evolve taking into account the time factor. Thus, it is one of the parameters that we include as a suggestion for future work. Regarding participants’ perspectives, this is mentioned (line 479) when they express their opinion about what they consider a healthy relationship in general, as well as the fundamental points for the relationship. However, the distinction between the characteristics of a healthy "casual" relationship vs healthy relationship "committed" could be addressed in complementary studies.